# Behavioural and antennal responses of *Aedes aegypti* (l.) (Diptera: Culicidae) gravid females to chemical cues from conspecific larvae

Antoine Boullis[1☯], Margaux Mulatier[1☯], Christelle Delannay[1], Lyza Héry[1], François Verheggen[2], Anubis Vega-Rúa[1]*

**1** Laboratory of Vector Control Research, Institute Pasteur of Guadeloupe–Lieu-dit Morne Jolivière, Les Abymes, Guadeloupe, France, **2** TERRA, Gembloux Agro-Bio Tech, University of Liège, Gembloux, Belgium

☯ These authors contributed equally to this work.
* avega@pasteur.fr

**Data Availability Statement:** All relevant data are within the manuscript and its Supporting Information files.

**Funding:** This work has been notably supported by the Programme Opérationnel FEDER-Guadeloupe-

## Abstract

Mass trapping of gravid females represents one promising strategy for the development of sustainable tools against *Aedes aegypti*. However, this technique requires the development of effective odorant lures that can compete with natural breeding sites. The presence of conspecific larvae has been shown to stimulate oviposition. Hence, we evaluated the role of four major molecules previously identified from *Ae. aegypti* larvae (isovaleric, myristoleic, myristic [i.e. tetradecanoic], and pentadecanoic acids) on the oviposition of conspecific females, as well as their olfactory perception to evaluate their range of detection. Using flight cage assays, the preference of gravid females to oviposit in water that previously contained larvae (LHW) or containing the four larval compounds was evaluated. Then, compounds and doses inducing the highest stimulation were challenged for their efficacy against LHW. Only isovaleric acid elicited antennal response, suggesting that the other compounds may act as taste cues. Pentadecanoic acid induced significant oviposition stimulation, especially when dosed at 10 ppm. Myristoleic acid and isovaleric acid deterred oviposition at 10 and 100 ppm, while no effect on oviposition was observed with myristic acid irrespectively of the dose tested. When the four compounds were pooled to mimic larvae's chemical signature, they favored oviposition at 1 ppm but negatively affected egg-laying at higher concentrations. When properly dosed, pentadecanoic acid and the blend of compounds may be promising lures for ovitraps as they could compete with LHW. Due to their low volatility, their effect should be further evaluated under field conditions, in addition with long-range attractants for developing effective tools against gravid females.

## Introduction

The yellow fever mosquito, *Aedes aegypti* (L.), is a serious threat for human health in tropical and subtropical regions due to its significant vectorial capacity for pathogens of medical importance [1, 2]. As no efficient prophylactic treatments are available against most of *Aedes*-borne diseases, vector control remains the most effective way to prevent and contain

Conseil Régional 2014–2020 (Grant 2018-FED-1084). LH is funded by a PhD scholarship from La Région Guadeloupe. AB is partially funded by a Calmette & Yersin postdoctoral fellowship from the Institut Pasteur Department of International Affairs. The funders had no role in study design, data collection and analysis, decision to publish, or preparation of the manuscript.

**Competing interests:** The authors have declared that no competing interests exist.

**Abbreviations:** UPW, Ultrapure water; LHW, Larvae-holding water; OAI, Oviposition activity index; EAG, Electroantennography; PPH, Preference-performance hypothesis.

outbreaks. During the past decades, vector control strategies heavily relied on the use of insecticides [3], contributing to the selection of multiple resistance mechanisms in vector populations [4], and leading to non-reversible effects on human health and on the environment, as seen elsewhere [5–9]. Consequently, there is an urgent need for developing alternative and sustainable control strategies that specifically target *Ae. aegypti*. In this context, the control of gravid females may be one promising route to limit population density without strong selective pressure (only a subset of the population would be targeted), as well as with a limited impact on the environment and on the non-target organisms [10]. Gravid females are also of special epidemiological interest because they have had at least one previous blood meal and are therefore more likely to be infected with arboviruses and involved in disease transmission. However, targeting this specific life-stage can only be achieved by an extensive understanding of their biology and most specifically of their oviposition behaviour.

As for several mosquito species, the selection of oviposition sites by *Ae. aegypti* gravid females is a key determinant for the survival and the optimal development of their progeny [11]. Gravid females rely on visual, olfactory and gustatory cues to assess water quality and to select the most suitable breeding site [12]. The chemical cues informing about the suitability of a breeding site can be sorted into different classes depending on the distance they act (e.g. long-, middle- or short-range) and the behavioural response they induce (e.g. attraction or repulsion, and oviposition stimulant or deterrent) [13]. These signals are known to originate from several sources (e.g. plant, microbial, conspecifics and heterospecifics) [13–15], with the presence of immature conspecifics being a strong determinant of the breeding site choice. Yet, laboratory experiments evidenced that eggs- and larval-holding waters strongly stimulate females for egg-laying, suggesting the presence of a pheromonal signal [16–20]. This preference is also observed under field conditions, where females preferred to lay eggs in containers that have previously held larvae or pupae [20–22]. The presence of immature stages, whose chemical signature is thought to originate from both larvae and their associated bacteria [13, 17, 23–25], is likely interpreted by females as a suitable site for larvae development [15]. On the other hand, larval density seems to be another determinant of the breeding site selection, as aversion has been observed under crowded conditions, which might be explained by female avoidance of detrimental competition [17, 18].

Thus, the increased ovipositional response in the presence of immature conspecifics offers a great potential in vector control, and the identification of the chemical compounds involved in the short and long-range attraction could lead to the development of oviposition lures that specifically target *Ae. aegypti* gravid females. For this purpose, a total of 13 carboxylic acids and corresponding methyl esters isolated from eggs have already been listed as influencing the oviposition of gravid females [13, 25]. Whether these compounds act at short or long range is yet to be elucidated. Also, an alkane identified from *Ae. aegypti* larvae [26] has been shown to influence the flight orientation of gravid females following a dose-dependent response [24]. More recently, the identification of 15 compounds (including 8 carboxylic acids, 2 corresponding methyl esters, and 1 lactone) in *Ae. aegypti* larval extracts represents to date the widest diversity of larval compounds identified [27], and raises questions about their implication in the oviposition behaviour. Most of them are long-chain fatty acids and are therefore expected to present low volatility and act as taste cues rather than at distance. However, despite these observations, the role of most of these compounds in mediating oviposition of gravid females as well as their sensory perception still remain to be investigated.

The present study gives more insight into the relationship between female oviposition behaviour and the chemical signature linked to the presence of immature conspecifics. Hence, we aimed to (*i*) confirm the impact of larval density on the oviposition site selection, (*ii*) assess the olfactory perception and the influence on oviposition behaviour of the four major

compounds identified from larval extracts by Wang and colleagues [27]: isovaleric acid ($C_5H_{10}O_2$), myristoleic acid ($C_{14}H_{26}O_2$), myristic (i.e. tetradecanoic) acid ($C_{14}H_{28}O_2$) and pentadecanoic acid ($C_{15}H_{30}O_2$), and to (*iii*) evaluate their competitiveness when compared to natural odours of *Ae. aegypti* immature stages. The data obtained provide clues to understand both the sensory perception of these compounds and their potential for being used in vector control.

## Materials and methods

### Ethic statement

The use of fresh human blood from healthy volunteers to feed mosquitoes was approved by the internal ethics committee of the Pasteur Institute of Guadeloupe, established since September 2015 (no agreement number for internal ethics board), after receipt of written informed consent from the participants.

### Mosquito colony

A metapopulation of *Ae. aegypti* was established by sampling around private houses where the residents gave their permission for mosquito larvae collection. The sampling activities were conducted from July to August 2019 in the 5 following localities of Guadeloupe (French West Indies): Les Abymes, Pointe-à-Pitre, Deshaies, Saint-François, and Anse Bertrand. Experiments were performed on the 4th and 5th generation of this colony. Mosquitoes were maintained under laboratory conditions of 26 ± 1°C and 40–60% RH. Larvae were reared at densities of 200–300 larvae / L in dechlorinated tap water and were fed rabbit pellets. Adults were given *ad libitum* access to a 10% sucrose solution. An artificial blood meal using a Hemotek feeding system (Hemotek Ltd.®; Blackburn, UK) and 4 ml of fresh blood from a healthy volunteer dispensed into 2 Hemotek feeders was provided to mosquitoes 7 to 10 days after emergence. After blood feeding, all individuals were cold-anesthetised and fully engorged females were visually sorted and maintained in cages with water source prior to the assays. A total of 12 engorgements were performed during all the experiments.

### Chemicals

Chemical compounds used for electroantennography (EAG) and oviposition bioassays were previously identified from extracts of immature stages of *Ae. aegypti* (3rd and 4th larval instars and pupae) [27]. The isovaleric acid, myristoleic acid, myristic acid and pentadecanoic acid synthetic compounds (purity ≥ 99%; Sigma Aldrich Inc., St-Louis, MO, USA) were diluted in *n*-hexane (HPLC grade; Carlo Erba reagents, Milano, Italy).

### Preparation of the chemical solutions

The four synthetic compounds were tested both individually and in blend at 4 concentrations. The blend was obtained by mimicking the proportions observed in larvae extracts as follows: 13% isovaleric acid, 53% myristoleic acid, 23% myristic acid and 11% pentadecanoic acid [27]. A second mixture of compounds was also prepared by removing myristoleic acid from the blend (respecting the proportions of the three other compounds, i.e. 28% isovaleric acid, 49% myristic acid and 23% pentadecanoic acid). After serial dilutions in *n*-hexane, 100 μL of preparation was subsequently added to 100 ml of ultrapure water (UPW) in the oviposition test bowl to obtain the required concentration. For all synthetic solutions, the 4 concentrations in oviposition bowls ranged from 0.1 to 100 ppm following a $\log_{10}$ increase, except for the mixture of compounds without myristoleic acid which was only tested at 1 ppm and 100 ppm.

Control bowls received 100 ml of UPW supplemented with 100 µl of solvent. The pH of the solutions was monitored by accredited standard methods (www.cofrac.fr) at the Laboratory of Environmental Hygiene at the Institute Pasteur of Guadeloupe to control for the influence of organic acids on solution acidity.

## Preparation of larval holding water (LHW)

Groups of 2nd and 3rd instar larvae were rinsed with UPW to avoid any remaining of food, and subsequently placed in glass cups containing 100 ml of UPW at 5, 20 or 100 larvae per cup. The density of 20 larvae / 100 ml corresponds to the optimal rearing density used as the standard in our facilities. The density of 5 larvae / 100 ml is used as a low density, whereas the density of 100 larvae / 100 ml is considered as a high density. The three different densities were selected because they are expected to induce quantitative differences in their larval-associated chemical signal. Larvae were maintained under the same laboratory conditions as for the rearing, but without food. After 3 days, water was filtered with fine stainless steel mesh to remove larvae and the remaining water (*i.e.* LHW) was used the same day for behavioural assays. At the time of filtration, the proportion of larvae in the water was the following: 25% 3rd instar, 50% 4th instar and 25% pupae. Control solution (*i.e.* UPW) was also filtered using stainless steel mesh to avoid any bias.

## Oviposition assays

Dual choice bioassays were carried out to measure the oviposition response of gravid *Ae. aegypti* females toward the tested solutions exactly 3 days after blood feeding. Two dark red ceramic bowls of 8 cm diameter were placed at opposite corners of a Bugdorm-1® test cage (30 × 30 × 30 cm; MegaView Science Education Services Co., Taiwan) and were filled with 100 ml of solution (treatment or control). A strip of filter paper (Whatman™, n° 2300 916) was partially immersed into each bowl to serve as oviposition substrate. For each replicate, a homogeneous group of 19 to 20 gravid females was released into the cage. After 24 h, bowls and papers were removed from the cage and eggs were visually counted under a binocular magnifier. Each trial (*i.e.* condition) was repeated 5 times, for which the position of the bowls in the cage was randomly attributed. Before each trial, bowls were soaked overnight in alkaline detergent (RBS T105; Chemical products R. Borghgraef, Brussels, Belgium), then abundantly rinsed and sterilized at 100°C for 1 h. All assays were performed at 26 ± 1°C and 60 ± 10% RH.

Three series of experiments were performed: first, to confirm the effect of larvae on oviposition site selection in our experimental set-up, the influence of LHW on the oviposition of gravid females was measured according to larval density; then, the oviposition response of gravid females toward the 4 selected compounds previously identified in extracts of *Ae. aegypti* immature stages was investigated; finally, the potential of these compounds to outperform LHW effect on oviposition was assessed. Experiments were performed as follows:

i. The influence of larval infusion at densities of 5, 20 and 100 larvae / 100 mL was tested against UPW.

ii. The influence of the selected compounds (individually or in blend) on oviposition preferences was tested at 0.1, 1, 10 and 100 ppm against UPW.

iii. Compounds and doses showing the strongest stimulant effect on oviposition were challenged against LHW at the density also showing the strongest stimulant effect.

## Electroantennography (EAG) assays

EAG assays were performed to assess the olfactory detection of the compounds tested in bioassays. To do so, a female mosquito was cold-anesthetised, after which the head was separated from the thorax. A glass capillary (1.35 mm OD, 0.95 mm ID; Hirschmann Laborgeräte GmbH, Eberstadt, Germany) previously filled with an electrolytic solution (NaCl 7.5 g/l, $CaCl_2$ 0.21 g/l, KCl, 0.35 g/l, $NaHCO_3$ 0.2 g/l) was placed into the posterior part of the head and connected to a reference electrode. The tips of both antennae were excised and connected to the recording electrode through a second glass capillary also filled with the electrolytic solution. The recording electrode was connected to an amplifier (IDAC-4; Syntech®, Hilversum, the Netherlands). A dose of 100 μg of compound was deposited (5 μl dosed at 20 μg.μl$^{-1}$) on a piece of filter paper (2 cm$^2$) placed into a glass Pasteur pipette. The solvent was allowed to evaporate for 10 s under purified airstream (600 ml.min$^{-1}$). Each stimulus was presented to the antennae using an air puff (0.3 s) introduced into a continuous humidified airflow (1200 ml.min$^{-1}$). The glass Pasteur pipette, the filter paper and the aliquot of solution deposited were renewed at each air puff. Time interval between two stimulations was 40 s. Antennal responses were digitized, amplified 10 times and processed using the software Autospike (V3.9; Syntech®). The antennal signal was filtered with a low cutoff set at 0.1 Hz. Gravid (9–12 days old) and post-oviposition (12–15 days old) females were tested in experiments (n = 10 for each group). Each individual mosquito was exposed to the 4 compounds as well as to the negative and positive controls (*n*-hexane and 1-octen-3-ol, respectively), presented following a random sequence.

In a second time, the detection threshold of the compounds that elicited antennal response at 100 μg was assessed. Five doses of a same compound were presented to the antennae, from $10^{-8}$ to $10^{-4}$ g following log$_{10}$ increments (plus positive and negative controls). Because no difference in antennal perception was observed between the two groups of females (i.e. before and after oviposition) in the previous tests, and to ensure consistency with the physiological state of females used in oviposition assays, only gravid females before oviposition (9–12 days old) were tested for this assay.

## Statistical analyses

All statistical analyses were performed using the software R 3.3.2 [28]. The mean oviposition activity index (OAI) [29], with values ranging from + 1 to– 1, was calculated for each trial as: OAI = ($N_T$−$N_C$) / ($N_T$ + $N_C$), where $N_T$ indicates the number of eggs laid in the treatment solution (LHW or water with synthetic compound(s)) and $N_C$ indicates the number of eggs laid in the control solution (UPW). In the third oviposition experiment, the treatment solution corresponds to the bowl containing the synthetic compound(s), whereas the control solution corresponds to LHW. For these assays, a positive OAI value indicates a preference toward the treatment solution, whereas a negative OAI value indicates aversion. Paired Student t tests were performed for each condition to test for a significant effect of the tested solution. The density-dependent effect of LHW on the OAI and the interaction between the presence of larvae and the density were evaluated using a linear model (lm function). For the oviposition assays involving synthetic compounds, the OAI was compared between compounds and doses using a linear mixed-effects model (lmer function, lme4 package, day of experiment coded as random factor), and the interaction between compound and dose was also tested. For both models, post-hoc comparisons were performed (Tukey's tests, multcomp package). Model selection was performed using AIC and analysis of the residuals (RVAideMemoire package), with non-significant interactions removed from the model.

EAG data was analysed by comparing the response of each compound with those from the solvent. After checking the conditions of application, a two-way analysis of variance

(ANOVA) was assessed to evaluate the impact of the compound and the gonotrophic status on the antennal response. Post-hoc comparisons were then performed to evaluate the antennal detection of each compound. Dose responses were analysed using linear mixed-effects models (lmer function, lme4 package, individual coded as random factor) and post-hoc comparisons between doses were performed (Tukey's tests, multcomp package).

## Results

### Larval-holding water promotes oviposition following a density-dependent effect

Larval holding water (LHW–i.e. water that previously contained larvae) stimulated gravid females for oviposition when compared to UPW in our experimental set-up, for all tested densities (Table 1). Also, a density-dependent effect was observed on the OAI ($F_{2,12}$ = 4.23, P = 0.04). Indeed, multiple comparisons evidenced significant differences between 5 larvae / 100 ml and 100 larvae / 100 mL, the latter density inducing the highest oviposition stimulation (Tukey's post-hoc; 5 versus 20 larvae / 100 mL: P = 0.91; 20 versus 100 larvae / 100 mL: P = 0.11; 5 versus 100 larvae / 100 mL: P = 0.049) (Fig 1).

### Larval-associated compounds modulate oviposition preferences

The Oviposition activity index (OAI) was significantly affected by the compounds tested ($\chi^2$ = 92.55, Df = 4, P < 0.001), the doses used ($\chi^2$ = 20.12, Df = 3, P < 0.001), with a significant interaction (compound × dose interaction: $\chi^2$ = 57.20, Df = 12, P < 0.001) (Fig 2). Positive OAI values were observed for pentadecanoic acid at doses of 1, 10 and 100 ppm, with attraction considered significant at 10 ppm when compared to UPW (OAI = 0.38 ± 0.09, Student's t-test: t = 4.12, Df = 4, P = 0.014). Conversely, OAI values for myristoleic acid were negative for all tested doses, with doses of 10 and 100 ppm eliciting significant deterrent effect (Student's t-tests;10 ppm: OAI = –0.53 ± 0.09, t = –3.95, Df = 4, P = 0.017; 100 ppm: OAI = –0.70 ± 0.07, t = –9.12, Df = 4, P < 0.001). OAI values for isovaleric acid were negative at doses of 10 and 100 ppm, but the differences with UPW were not significant (Student's t-tests, P > 0.05 for all doses). Also, myristic acid did not elicit significant effect on oviposition, with constant OAI values among doses, ranging from -0.11 to 0.10 (Student t-tests, P > 0.05 for all doses). The blend of compounds induced greater variability on the OAI among doses, with significant stimulation at 1 ppm (OAI = 0.21 ± 0.04, Student t-test: P = 0.002) and significant deterrence at 100 ppm (OAI = -0.65 ± 0.08, Student t-test: P < 0.001). The doses of 0.1 ppm and 10 ppm did not induce significant effect on oviposition (Student t-tests, P > 0.05 for these doses). To investigate whether the presence of myristoleic acid reduced the attractiveness of the blend, the solution was tested by removing this compound at 1 ppm and at 100 ppm. Such doses were selected because a significant influence on oviposition was previously observed with the full blend. When the blend was tested without myristoleic acid, similar results were observed. Females preferred to oviposit in the bowl containing this solution dosed at 1 ppm

**Table 1. Oviposition responses (number of eggs laid) of *Ae. aegypti* gravid females towards larval holding water (LHW) at different densities compared to ultrapure water (UPW).**

| Larval density | Number of eggs laid (Mean ± S.E.M.) | | F stat (Student t-test) | P-value |
|---|---|---|---|---|
| (larvae / 100 mL) | LHW | UPW | Df = 4 | |
| 5 | 410 ± 34 | 241 ± 32 | 6.73 | 0.002 |
| 20 | 408 ± 48 | 221 ± 30 | 2.92 | 0.043 |
| 100 | 583 ± 63 | 193 ± 24 | 6.51 | 0.002 |

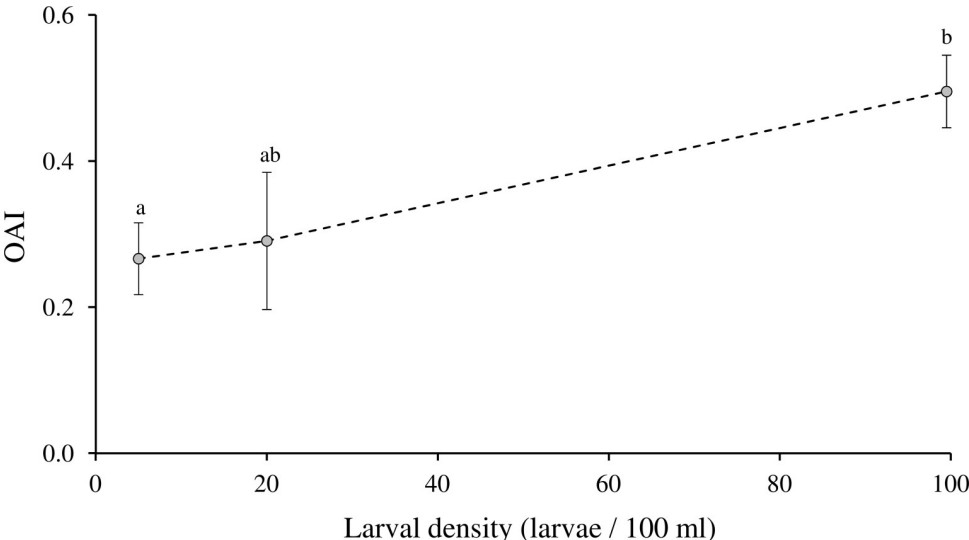

**Fig 1. Oviposition activity index (OAI) of *Ae. aegypti* gravid females towards larval holding water (LHW) at three different densities: 5, 20 and 100 larvae / mL (mean ± S.E.M., n = 5 replicates per density).** OAI value of 0 indicates no difference in oviposition between LWH and UPW bowls. Different letters indicate significant differences between densities (Tukey's post-hoc test: P < 0.05).

over UPW, but without significant difference (OAI = 0.18 ± 0.07, Student t-test: P = 0.068), whereas they displayed significant avoidance when this solution was dosed at 100 ppm (OAI = -0.21 ± 0.07, Student t-test: P = 0.048) (Fig 2). The pH measurements conducted showed that only isovaleric acid induced a strong acidification of the solution at 100 ppm (pH = 3.36) when compared to UPW (pH = 5.35) (see S1 File). However, this compound did not induce significant difference in the attractiveness compared to UPW at this dose (Fig 2).

## Two of the larval-associated compounds tested may act as taste cues

The antennal detection of the four carboxylic acids was assessed in gravid *Ae. aegypti* females 3 to 6 days after their blood meal. For all compounds, the oviposition status of females (*i.e.* before and after oviposition) did not influence perception by the antennal olfactory apparatus ($F_{1,90}$ = 0.04, P = 0.85). Among the compounds that influenced *Ae. aegypti* oviposition, myristoleic acid and pentadecanoic acid showed no significant differences in antennal detection when compared to the hexane solvent (Tukey's post-hoc comparisons with solvent: myristoleic acid: P = 0.99; pentadecanoic acid: P = 0.98), while isovaleric acid elicited a significant antennal response (Tukey's post-hoc comparisons with solvent: P < 0.001) (Fig 3). The antennal detection threshold for isovaleric acid was observed at $10^{-5}$ g (Tukey's post-hoc comparisons with solvent: $10^{-8}$ to $10^{-6}$ g: P > 0.05; $10^{-5}$ and $10^{-4}$ g: P < 0.001) (Fig 4). Regarding myristic acid (for which no influence on oviposition was evidenced), the antennal detection was not significant when compared to the solvent (Tukey's post-hoc comparisons with solvent: myristic acid: P = 0.99) (Fig 3).

## Synthetic compounds are competitive toward natural larval signature

The compound and dose that elicited the best stimulant effect in oviposition assays were challenged against LHW at larval concentration showing the highest OAI. Therefore, pentadecanoic acid dosed at 10 ppm and the blend of compounds dosed at 1 ppm were individually challenged against LHW at density of 100 larvae / 100 mL. In both oviposition tests, no

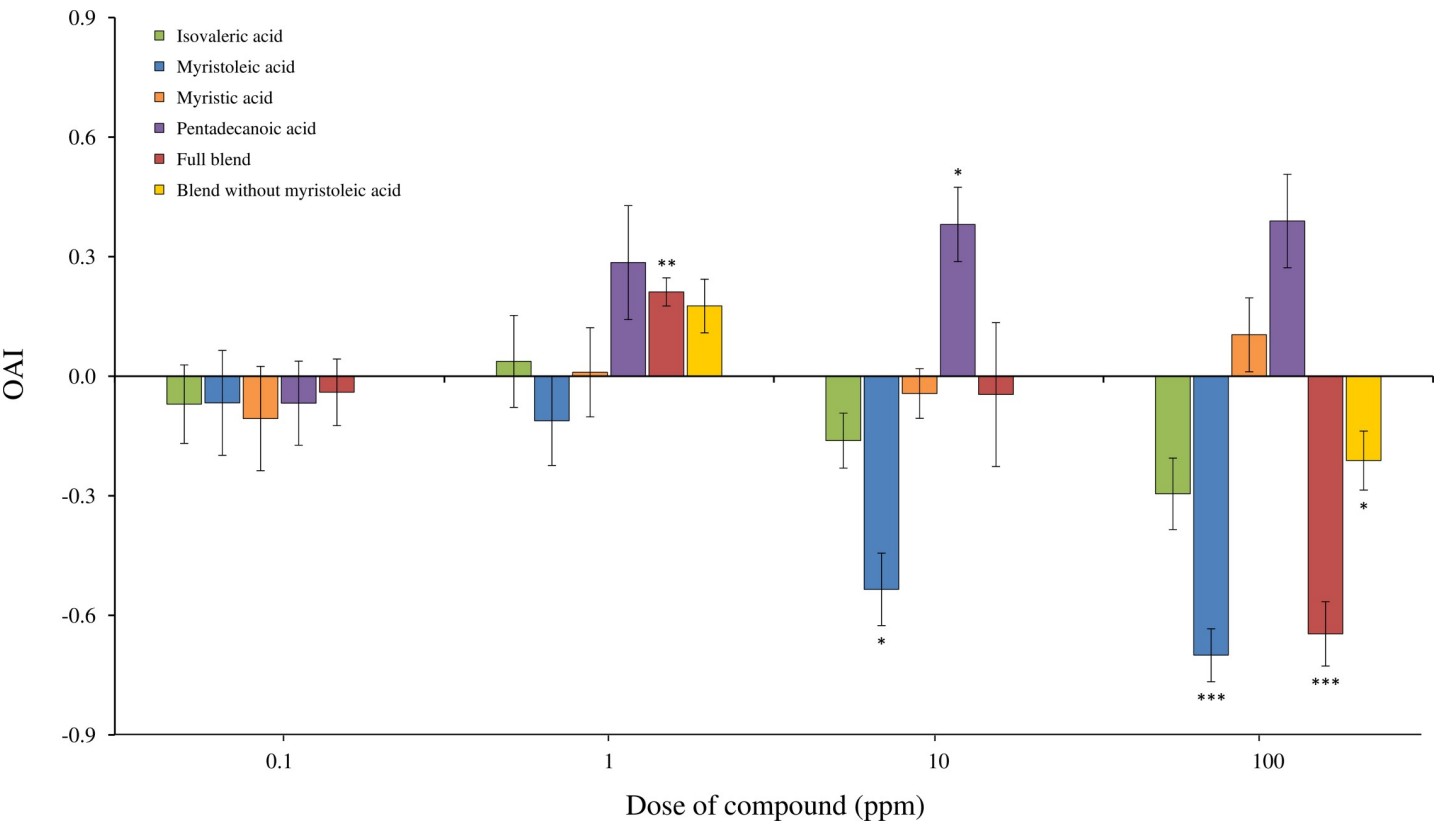

**Fig 2. Oviposition activity index (OAI) of *Ae. aegypti* gravid females towards the different synthetic solutions at four different doses: 0.1; 1; 10 and 100 ppm (mean ± S.E.M., n = 5 replicates per condition).** Asterisks show significant effect of the condition over water (paired Student's t.test: * P < 0.05, ** P < 0.01, *** P < 0.001).

significant differences were observed in the preferences between the synthetic mixture and LHW (Table 2).

## Discussion

The selection of breeding sites by mosquitoes can be explained by the preference-performance hypothesis (PPH) [30] and depends on multiple factors, such as the presence and density of conspecifics, the presence of natural enemies, and the abundance of nutrients [31]. The data presented here confirm the positive influence of conspecific immature stages on the oviposition of gravid *Ae. aegypti* females, as already observed in literature [16–20]. The presence of larval-associated signals within a breeding site may notify gravid females about suitable conditions that can ensure the proper development of their progeny. We also highlighted a positive correlation between larval density and oviposition preference, with OAI values reaching +0.50 under a density of 1000 larvae / L. However, significant differences were observed only between the lowest and the highest tested densities, and not between each tested ones, suggesting that the chosen larval quantities are not discriminating enough to observe a substantial difference in the amount of chemical cues emitted between two successive densities in our experiments. The observed correlation between density and attractiveness supports the results from previous studies on the same mosquito species, where a strong preference was observed with densities of the same order of magnitude (OAI about +0.60 for densities of 1000 to 2000 larvae / L) [17, 32]. Furthermore, in both studies, a decrease in oviposition preference towards

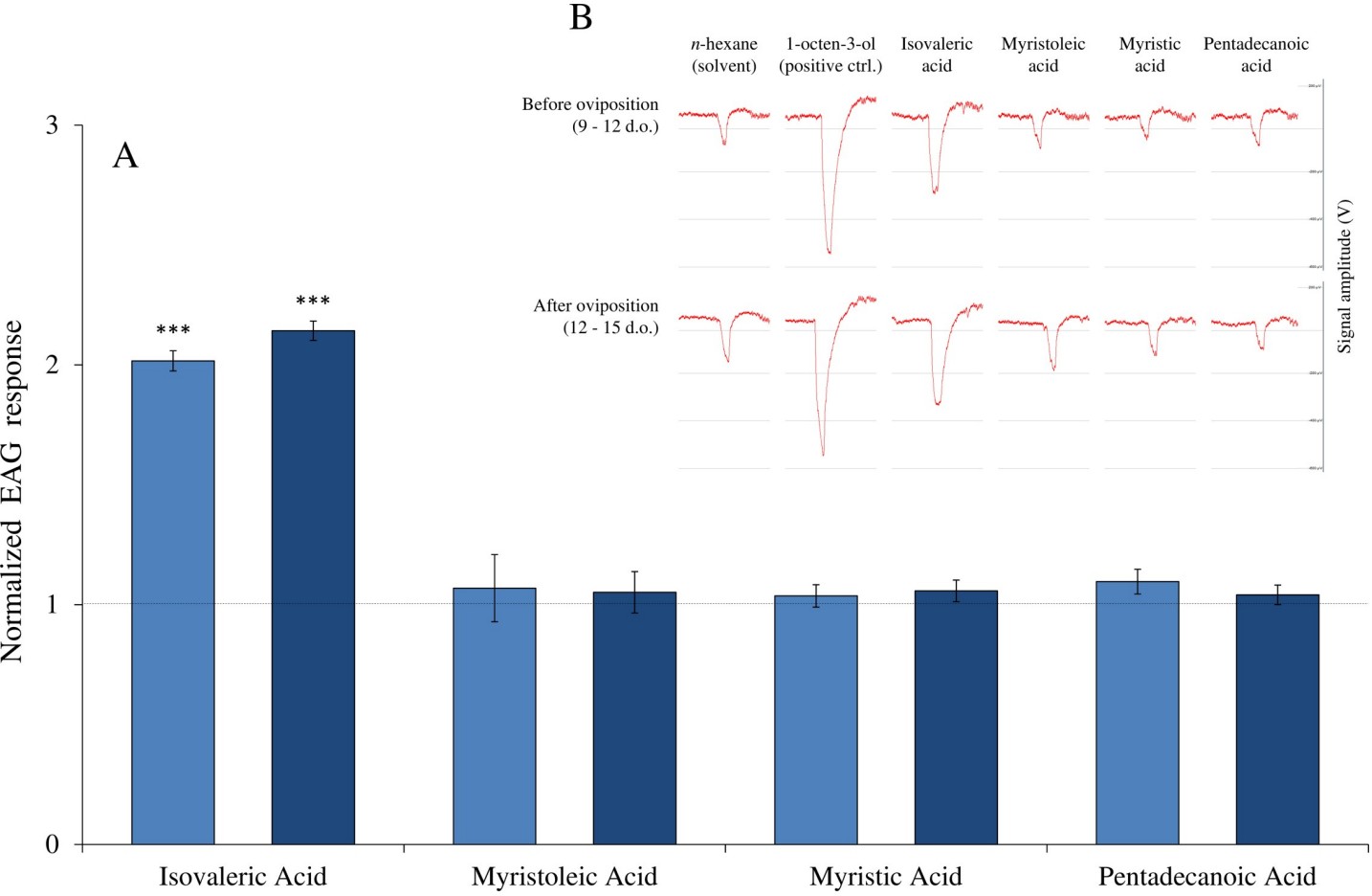

**Fig 3. A**: Normalized antennal responses of engorged *Aedes aegypti* females towards synthetic compounds at $10^{-4}$ g (mean ± S.E.M., n = 10 individuals per group). Normalized EAG values correspond to the response to the compound divided by the response to the solvent (i.e. hexane). Light blue bars correspond to females before oviposition (9–12 days old); dark blue bars correspond to females after oviposition (12–15 days old). Asterisks indicate significant differences in the antennal perception compared to the solvent (Tukey's post-hoc test: *** P < 0.001). **B**: Exemplar traces obtained from Autospike® software for the two groups of mosquitoes toward the different test solutions.

LHW was observed with higher larval densities (4000 to 7800 larvae / L) [17, 32] but we did not reach this aversive threshold in our experiment. Through changes in the concentration of larval-associated signals, larval density may inform females about the suitability of the breeding site, either by stimulating oviposition until the optimal larval concentration, or by inducing deterrence when larval density represents a risk of competition.

Given the positive influence of immature stages on the oviposition of *Ae. aegypti* gravid females, the identification of their chemical signature has received increasing interest from the scientific community. In this respect, several organic compounds, mainly carboxylic acids, have been identified from immature stages and have been shown to modulate the choice of gravid females, following either attractive/stimulant or repellent/deterrent effects [13]. For instance, two carboxylic acids recently identified in larval extracts, dodecanoic acid and tetradecanoic (i.e. myristic) acid [27], have also been previously found in *Ae. aegypti* eggs extracts [13]. They have been shown to significantly enhance the oviposition of *Ae. aegypti* females in both laboratory and semi-field conditions [23, 33]. To further investigate the influence of carboxylic acids on *Ae. aegypti* gravid females, we tested the four major compounds identified in larval extracts by Wang and colleagues [27] for their role in mediating their oviposition

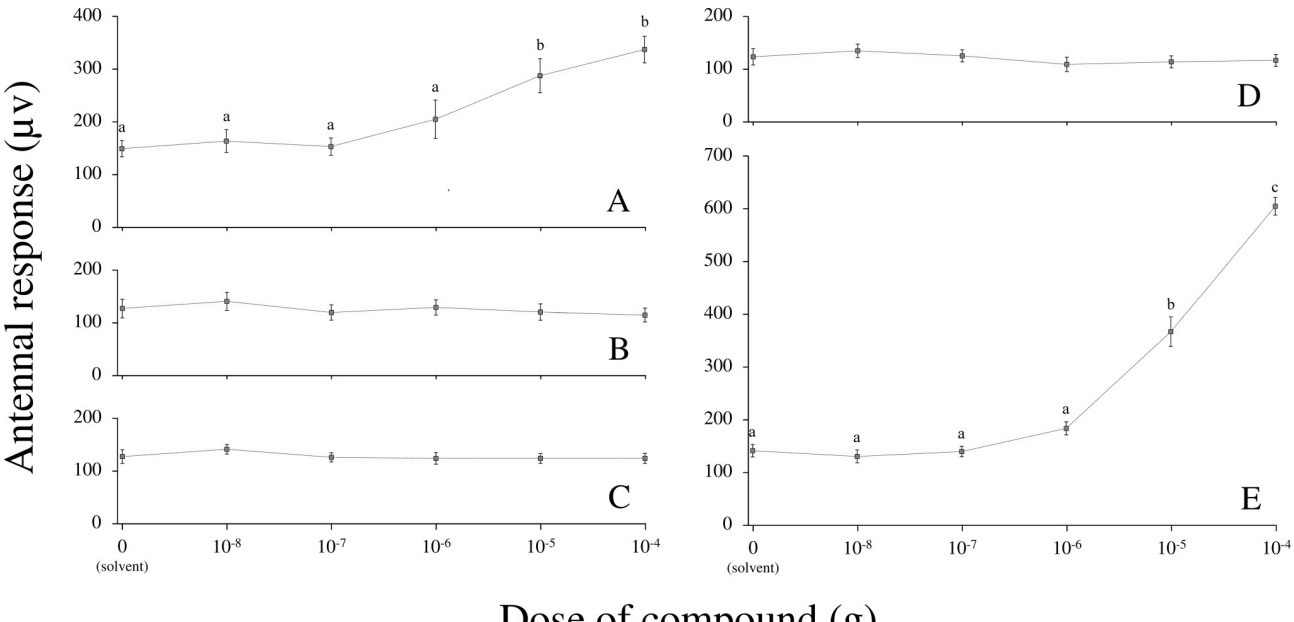

**Fig 4. Antennal dose-responses of engorged *Aedes aegypti* females towards the four carboxylic acids and the positive control at 5 different doses, from $10^{-8}$ to $10^{-4}$ g following a $\log_{10}$ increase (mean ± S.E.M.). A:** Isovaleric acid (n = 6). **B:** Myristoleic acid (n = 5). **C:** Myristic (= tetradecanoic) acid (n = 5). **D:** Pentadecanoic acid (n = 5). **E:** 1-octen-3-ol (positive control, n = 7). Different lowercase letters indicate significant differences in perception between doses within a compound (Tukey's post-hoc: P < 0.05).

behaviour. These compounds, isovaleric acid, myristoleic acid, myristic acid and pentadecanoic acid, were found to be specific markers of late developmental stages of immature *Ae. aegypti*: they are present in significant quantities in $L_4$ larvae and pupae, except isovaleric acid which is present in $L_4$ larvae but absent from pupae [27].

Our results evidenced complex and contrasted oviposition responses elicited by the four tested carboxylic acids. First, pentadecanoic acid strongly stimulated females for oviposition, with a positive correlation between dose and OAI. This compound has been identified for the first time in *Ae. aegypti* immature stages [27] and its effect on oviposition had never been assessed before. Interestingly, pentadecanoic acid is also present in human skin emanations [34] and has been found to be repellent in olfactometer assays towards host-seeking *Ae. albopictus* (Skuse) females [35]. In this study, we did not find any antennal response to this compound, suggesting that, due to its low volatility, it may serve as a taste cue for gravid females once they land on the water surface. It is not excluded that, as for the others compounds where no antennal response was observed, others sensory organs such as maxillary palps, proboscis, or tarsi might be involved in this detection. Therefore, its behavioural effect seems to be strongly dependent upon the mosquito species and physiological status considered.

**Table 2. Oviposition responses of *Ae. aegypti* gravid females towards synthetic compounds compared to LHW at 100 larvae / 100 mL (OAI: Oviposition activity index calculated towards the synthetic compounds).**

| Compound | Dose | Number of eggs laid (Mean ± S.E.M.) | | OAI | Statistics [a] | P-value |
|---|---|---|---|---|---|---|
| | | Treated water | LWH | (Mean ± S.E.M.) | | |
| Pentadecanoic acid | 10 ppm | 223 ± 42 | 204 ± 60 | 0.07 ± 0.22 | t = 0.2 | 0.85 |
| Blend | 1 ppm | 222 ± 56 | 332 ± 69 | -0.19 ± 0.14 | V = 2 | 0.19 |

[a] t: Student paired t.test; V: Wiloxon Mann-Whitney paired test

Isovaleric acid and myristoleic acid induced negative oviposition responses, especially when dosed at 10 and 100 ppm. These compounds might indicate to gravid females an unsuitable site for oviposition, such as a resource-depleted breeding site [36], or potentially inform them about the presence of late instar stages and thus of a risk of cannibalism for the young progeny [37]. This phenomenon has been recently described in *Anopheles coluzzii* Coetzee & Wilkerson, where the presence of $L_4$ larvae had a negative effect on the oviposition of conspecific gravid females [38]. Myristoleic acid has already been identified in *Ae. aegypti* eggs as a minor compound [25], whereas in $L_4$ larvae and pupae it is the predominant one (32 to 40% of the whole chemical signature) [27]. Despite such high abundance in late immature instars, the effect of this compound on mosquito oviposition had never been assessed before. Myristoleic acid did not elicit antennal response in our experimental set-up. Due to its chemical nature, it is expected to present low volatility and may rather act as a taste cue than at long distance. Concerning isovaleric acid, its deterrent effect was only observed at high concentrations. When tested at 100 ppm, although the OAI reached the deterrent threshold of –0.30 defined by Kramer & Mulla [29], its effect was not statistically significant. Also, this compound has been found in human sweat [39] and in larval food infusion [40], probably as a result of bacterial metabolism [41]. Hence, when highly dosed, isovaleric acid may inform females about a high bacterial activity within a breeding site, unsuitable for larval development. This compound may also induce contrasting behaviours depending on the mosquito species and physiological status considered. For example, isovaleric acid has been shown to deter oviposition of *Culex quinquefasciatus* Say at 600 ppm [42], but its addition to a volatile blend of ammonia and lactic acid enhanced the attraction to host-seeking *An. gambiae s.s.* Giles females [43]. Also, isovaleric acid is the only tested compound that elicited antennal responses in our electrophysiological assays. Further behavioural studies are thus needed to better understand the range of action of this compound (i.e. long or middle/short distance), as well as its influence on other physiological groups (e.g. host-seeking females).

Myristic acid did not significantly influence oviposition regardless of the tested dose and was not detected by the antennal olfactory apparatus. This is contrasting with previous bioassays demonstrating a highly stimulant effect of this compound at doses below 10 ppm [23, 44, 45]. These discrepancies demonstrate the complex oviposition response of *Ae. aegypti* towards synthetic compounds, which might be influenced by the mosquito strain tested, its laboratory colonization history [31], mosquito individual experience [46], and the laboratory conditions used for bioassays.

The blend of compounds favoured oviposition at 1 ppm and had a deterrent effect at 10 and 100 ppm. This tendency is in accordance with the density-dependent trend observed with larval holding water. Yet, a dose of 1 ppm may represent an attractive density of larvae and thus a suitable breeding site, whereas higher doses can mimic crowding conditions and a risk for competition. Interestingly, myristoleic acid was found to be the main component of the blend (53%) and, even combined with the stimulant pentadecanoic acid, maintained its deterrent activity when the blend was dosed at 100 ppm. When myristoleic acid was removed from the blend at 100 ppm, this combination maintained its repellence, although at a lower degree. This observation may be due to the presence of isovaleric acid in the blend, which showed a repellent effect when highly dosed. Yet, under high concentrations, the repulsive compounds within the blend might have a strong influence on oviposition. Besides, when dosed at 1 ppm, the full blend contained 0.11 ppm of pentadecanoic acid and significantly stimulated oviposition, whereas pentadecanoic acid presented individually at the dose of 0.1 ppm did not induce any behavioural effect. This highlights the ecological importance of testing molecules together by respecting their proportions, in order to obtain synergistic effects such as those observed with natural stimulants. Also, testing the blend without myristoleic acid at this dose did not

increase the oviposition preference of this solution toward gravid females, supporting here again the hypothesis of a synergistic effect between some of these compounds. Testing molecules as a blend also allows to take into consideration their specific effects depending on mosquito physiological state (for example host-seeking *versus* gravid females), and to better target the stage of interest. Indeed, many carboxylic acids identified from *Ae. aegypti* immature stages are also present in human skin emanations [34, 39], and their influence on mosquitoes might change depending on the other compounds they are associated with. Finally, pH values obtained from the solutions did not evidence any correlation between acidification of the solutions due to the addition of organic acids and mosquito oviposition choice. This suggests that the influence of these compounds on oviposition behaviour is driven by the chemical nature of the compounds.

The influence of larvae-associated chemicals on *Ae. aegypti* oviposition offers new perspectives for the control of gravid females. Gravid *Ae. aegypti* females are good targets for vector control as *(i)* they are more susceptible to carry pathogens, and *(ii)* their control may allow to reduce their progeny and therefore population densities [47]. For this matter, it is crucial to develop robust and specific vector control tools, as well as to compare their performance against naturally attractive breeding sites. Here we identified two oviposition stimulants, considered as good candidates for the development of ovitraps or gravitraps, by their ability to challenge attraction of natural breeding sites with larval chemical signatures: pentadecanoic acid dosed at 10 ppm and the blend of compounds dosed at 1 ppm. They may represent promising attractive lures, even if the blend contains compounds that are deterrent when tested individually. One compound indeed, myristoleic acid, triggered deterrence in our assays and could also be a good candidate for the development of repulsive odour-based lures. As these compounds are not detected by the antennal olfactory apparatus, they could be used in association with volatile cues in control programs. For instance, some plant extracts have been found to affect gravid females at long distance [13], and their association with signals from immature mosquito conspecifics may also increase the influence on females when they seek or land on potential oviposition sites, as well as improve the specificity to the targeted organisms. In an ovitrap-based system, the association of attractants with biocide agents may improve the efficiency of vector control, as demonstrated with an association of caproic acid and Temephos [25], as well as with LHW with *Bti* [20]. However, to drastically reduce the potential effect of biocides on non-target organisms, it is crucial to develop lures that specifically target *Ae. aegypti*. In this prospect, an accurate association of criteria (physical and chemical aspects) is needed.

It is noteworthy that the challenging effect of a synthetic attractant observed in laboratory compared to natural breeding sites might not necessarily be obtained under field bioassays. Firstly, *Ae. aegypti* females exhibit a "skip-oviposition" behaviour and tend to distribute eggs in several breeding sites [48], meaning that even an attractive site will not receive all female eggs. Secondly, others factors may interfere with the attraction of gravid females [15], especially the bacterial composition and their associated semiochemicals [44]. It is thus of paramount importance to depict the relative efficacy of candidate attractants when compared with naturally attractive breeding sites under field conditions. In addition to the attractants optimization, the removal of competing water-holding containers as recommended by Johnson and colleagues [47] seems to be a major asset to improve vector control campaigns involving ovitraps and/or gravitraps.

## Conclusion

In this study, we demonstrated the role of recently identified *Ae. aegypti* larval compounds on the oviposition of gravid females, and provided evidence about their perception by the

antennal olfactory apparatus. In a context of vector control, two main stimulants were identified as synthetic lures, having the potential to challenge natural odours of immature *Ae. aegypti* stages. One compound was also evidenced as a good oviposition deterrent. Given that these lures most likely act as taste cues for gravid females, this study addresses some insights about how to optimally use them in vector control strategies. Further studies are needed to establish their efficacy under field conditions and their possible combination with others molecules (for long-range attraction or biocidal effect), as well as to evaluate the specificity of these lures with regards to non-target species. Under a more theoretical prospect, this study gives insights about the sensory detection of larvae-associated chemicals of behavioural importance, and highlights the need for using combinations of chemical cues, taking their ratio into consideration, to better mimic the natural signals and to obtain behavioural responses as close as those elicited by natural cues.

## Supporting information

**S1 File. Study raw data.**
(XLSX)

## Acknowledgments

Authors thank the Master students Hortense Smith and Caitlin Gaete who contributed to the experiments.

## Author Contributions

**Conceptualization:** Antoine Boullis, Anubis Vega-Rúa.

**Data curation:** Antoine Boullis, Margaux Mulatier.

**Formal analysis:** Antoine Boullis, Margaux Mulatier, Anubis Vega-Rúa.

**Funding acquisition:** Anubis Vega-Rúa.

**Investigation:** Antoine Boullis, Margaux Mulatier, Christelle Delannay, Lyza Héry, Anubis Vega-Rúa.

**Methodology:** Antoine Boullis, Margaux Mulatier, Christelle Delannay, Anubis Vega-Rúa.

**Resources:** Antoine Boullis, Christelle Delannay, François Verheggen, Anubis Vega-Rúa.

**Supervision:** François Verheggen.

**Validation:** Antoine Boullis, Margaux Mulatier, Anubis Vega-Rúa.

**Visualization:** Antoine Boullis, Margaux Mulatier, Anubis Vega-Rúa.

**Writing – original draft:** Antoine Boullis, Margaux Mulatier, Anubis Vega-Rúa.

**Writing – review & editing:** Antoine Boullis, Margaux Mulatier, Lyza Héry, François Verheggen, Anubis Vega-Rúa.

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
