## [Decision Letter · Decision Letter 0]

1 Dec 2020

PONE-D-20-33407

Behavioral and antennal responses of Aedes aegypti (L.) (Diptera: Culicidae) gravid females to chemical cues from conspecific larvae

PLOS ONE

Dear Dr. Vega-Rúa,

Thank you for submitting your manuscript to PLOS ONE. After careful consideration, we feel that it has merit but does not fully meet PLOS ONE’s publication criteria as it currently stands. Therefore, we invite you to submit a revised version of the manuscript that addresses the points raised during the review process.

I agree with the points raised by the reviewers and find the following points deserve your attention:

Lines 87-88 Not sure of what you mean by  “interplay.”

Lines 172-173 Repolarization is very fast (<sec). There is probably a confusion with "sensory adaptation" which is a distinct phenomena

Lines 337-339 Was the pH of the solutions monitored? Addition of large amounts of organic acids are expected to decrease the pH, which could affect behavior.

Line 404 Typng errors: electroantennography instead of elactro; performance instead of performance

Line 406. Please give their names

We look forward to receiving your revised manuscript.

Kind regards,

Michel Renou, Ph.D

Academic Editor

PLOS ONE

Journal Requirements:

2. In your Methods section, please provide additional location information of the sampling sites, including geographic coordinates for the data set if available.

3. In your Methods section, please provide additional information regarding the permits you obtained for the work. Please ensure you have included the full name of the authority that approved the sampling sites access and, if no permits were required, a brief statement explaining why.

Reviewers' comments:

Reviewer's Responses to Questions

**Comments to the Author**

1. Is the manuscript technically sound, and do the data support the conclusions?

Reviewer #1: Partly

Reviewer #2: Yes

Reviewer #3: Yes

2. Has the statistical analysis been performed appropriately and rigorously? 

Reviewer #1: Yes

Reviewer #2: N/A

Reviewer #3: N/A

3. Have the authors made all data underlying the findings in their manuscript fully available?

Reviewer #1: Yes

Reviewer #2: Yes

Reviewer #3: No

4. Is the manuscript presented in an intelligible fashion and written in standard English?

Reviewer #1: Yes

Reviewer #2: Yes

Reviewer #3: Yes

5. Review Comments to the Author

Reviewer #1: In this study, Boullis et al., describe the effects of larval-associated carboxylic acids on the oviposition behavior and olfactory detection of gravid female Aedes aegypti. Using a dual choice-bioassay, they provide evidence that individual compounds have distinct effects and that some of these effects are dose-dependent. Moreover, they show that the odor blend is more attractive than the control at the lowest dose. Finally, antennal responses were tested using the EAG technique, showing response to isovaleric acid. In general, the data presented in this study are interesting and valuable but appear incomplete.

Strengths

The combination of sensory and behavioral techniques are suitable to explore the effect of carboxylic acids on oviposition. The authors’ rationale is based on published evidence that larvae, which release carboxylic acids, attract gravid females.

Weaknesses

Although others have looked at the behavioral effect elicited by 1,000-7,800 larvae, showing more data points (5) for the OAI would be desirable to show a more convincing dose-response relationship in the context of the authors’ bioassay. Two of the three data points (0 and 20 larvae/100mL) shown are not different.

The results shown in Table 2 (no difference between compounds and LWH) can also be explained by the mixing of both headspace within the 27L container.

What was the composition of the gravid females used in the bioassay experiments? In other words, were all gravid females controlled for the time post-blood feeding? The longer females hold laying eggs the more likely they are to lay eggs in any moist surfaces. This could profoundly confound the experiment outcomes.

Skip-oviposition (line 377) may also influence the outcomes of the behavioral experiments. It is hard to gauge this effect in the mosquito strain used here since the authors did not dissect the ovaries of the females after the experiments.

Minor comments

Replace x-axis legend with “Isovaleric acid (g)”. Is this a log [dose]?

Lines 24-25. “Only isovaleric acid elicited antennal response, suggesting that the other compounds may act as tactile cues.” The authors should discuss the possibility that the other odorants are detected by the maxillary palps or the proboscis.

Reference 3, Italicize species name.

Please provide more detailed description of the test cages.

Reviewer #2: - Abstract: The conclusion ” Pentadecanoic acid and the blend of compounds are promising lures for ovitraps as they could compete with LWH.” This is not 100% accurate as two compounds within the blend function as deterrent. I would suggest to re-word that phrase.

- Line 30-31:

It states that the blend of compounds is a promising lure for ovitraps as they can compete with LHW. However, results showed that two of the compounds function as deterrent. Perhaps it is better to state that some of the compounds could function as lures given that they increased oviposition.

- Line 90-92

Why did you choose only these four compounds? Wang and collegues (27) identified 8 compounds that were present in L4 larvae.

- Line 98: Spelling mistake: feed instead of fed

- Line 104: Please specify the five localities. This information is important in case another group wants to re-do the experiments.

- Line 109: Please specify the volume of blood that was used for feeding the mosquitoes.

- Line 110: There is no information on how you determined that the females were gravid.

- Line 124: Please specify the final concentration in oviposition bowls.

- Line 128: Please specify why you used these three different densities.

- Line 138: Advice for any future oviposition assay: Use either a bigger cage or smaller oviposition vessels. The oviposition vessels were quite large and this can have an olfactory effect between test and control in such a small cage thus causing an effect on the results.

- Line 141: 24h is a short amount of time for the females to choose an oviposition site and complete oviposition. If you have based this time period on a previous study please specify.

- Line 172: Please specify how often the filter was changed and new aliquots were added.

- Line 181: Why did you not have the same number of repetitions as in previous EAG experiment?

- Table 1: There is not significant difference between the densities 5 and 20, could you please discuss this further? Also the different between 20 and 100 is only approximately 180 eggs. How would you explain these results?

- Line 219: “Myristoleic acid exhibited lower OAI compared to other compounds”. This sentence is unnecessary as it is clear that it is negative across all concentrations and also it is explained in Line 224.

- Line 222: I don’t quite understand where the comparison between each compound and UPW is shown. There is no bar in Figure 1 that shows UPW (which I am assuming is the control).

- Line 227: Another comparison is made between compound and UPW however, the data is not presented in Fig. 2.

- Line 230-232: Why did you include the compounds that act as a deterrent in the blend? I would suggest to add another oviposition experiment where you remove these compounds and test the ones that gave a positive oviposition effect. Also add them according to respective ratios. This might give a more conclusive result and also strengthen the conclusion that they may be a good candidates as lures in the field.

- Line 236: How come recently engorged females were used? It takes females approximately 48hrs to process the nutrients present in the blood-meal, it is only after this they actively seek after an oviposition site.

- Line 332: It is strange that myristic acid did not have any influence on oviposition even though it has clearly been shown in ref(23,44,45). Did you try the concentration that was tried in these previous studies?

- Figure 1: The graph is not well explained (in the text or in the figure legend). Is my interpretation correct?: That a density of 5 is significant different from 100 but not 5 from 20 or 20 from 100? Please clarify the results. Perhaps use bars instead of a linear graph?

- Figure 2: Control is not presented in the graph. The main text keeps mentioning the comparison between compounds and UPW but this is not presented in the graph.

- Figure 3: Females seem to give a stronger response towards isovaleric acid post- oviposition. Why do you think this is? Also is it significant?

Reviewer #3: This study by Bouillis et al. focuses on the responses of Aedes aegypti female mosquitoes to cues associated with the presence of conspecifics larvae. More specifically the authors focused on 4 acids previously identified as chemicals produced by larvae. The main aim of the study was to determine the valence of these chemicals for females before and after oviposition. Due to the several deadly pathogens that Ae. aegypti females can transmit, it is essential to explore new avenues for vector control and targeting oviposition behavior is highly relevant.

The paper is clear, well written and rich in references. Data analyses are overall well conducted. Yet, the quality of the figures could be improved. I particularly value the fact that the mosquitoes used for the experiments are from a recently established colony which reduces the risk of genetic drift. One of my main concerns is that the reader does not have access to all the data the authors are mentioning so it is somehow difficult to assess their results which unfortunately affects the manuscript quality.

Specific comments:

- L132: Did you observe mortality in your larvae / pupae groups? I am wondering if dead larvae would influence the water odor profile.

- L137: what color were the bowls?

- L141: did you control for female size / weigh?

- L155-156: please rephrase. Currently sounds like you tested UPW at 0.1, 1, 10 and 100 ppm.

- L172: I suggest humidifying the airflow for acquiring EAG data.

- L173: “amplified”: please provide a value here. Also were your data filtered?

- L174: were the females starved from sucrose before performing the EAGs?

- L180: why not testing post-oviposition females as well? What about mated females but not blood-fed? The physiological status is expected to influence responses to odorants.

- L187: Student t tests: did you apply a Bonferroni correction for multiple comparisons?

- L200: conducting an ANOVA with such a small sample size is not appropriate.

- L285: tetradecanoic acid = myristic acid. This should be mentioned in the introduction.

- Figure 2. The blend was tested for the OAI, showing an attraction at 1 ppm. Why not testing it with EAGs? It would be interesting to test the 4 concentrations used for these oviposition experiments (i.e., 0.1; 1; 10; 100 ppm). Indeed, the combination of chemicals might trigger a higher antennal response, as it has been shown in mosquitoes and many other insect species.

- Figure 3. Please provide exemplar EAG traces for each condition.

Given that the conditions are independent (gravid or after oviposition), a space should be added between the bars.

It would be great to see the raw data and how responses to acids differ from responses to your positive control, octenol.

- Figure S1 should be included in the main paper instead of being provided as supplementary information.

Moreover, for transparency, please include the data obtained for the 3 other tested acids along with the responses obtained for the positive control (octenol) and for the solvent.

I suggest creating a panel highlighting all these data within one figure.

Why not including dose response curves performed with females after oviposition as well? It would be interesting to see if the threshold of detection is affected by the physiological status of the females.

- “Contact cues” or “tactile cues” are mentioned several times in the paper. I would replace it with “taste cues” for more accuracy.

- Please provide page numbers in your revised manuscript.

6. PLOS authors have the option to publish the peer review history of their article (what does this mean?). If published, this will include your full peer review and any attached files.

Reviewer #1: **Yes: **Jonathan D. Bohbot

Reviewer #2: No

Reviewer #3: No

---

## [Author Response · Author response to Decision Letter 0]

15 Jan 2021

Response to Reviewers

Title: Behavioural and antennal responses of Aedes aegypti (L.) (Diptera: Culicidae) gravid

females to chemical cues from conspecific larvae

Journal: PLoS One

Dear Pr. Renou,

On behalf of all co-authors, I would like to thank you as well as the three reviewers for the constructive remarks that contributed to improve the quality of our manuscript.

Two versions of the revised manuscript amended following the reviewer’s instructions were provided on the online submission platform. In the “marked-up version”, our changes are highlighted in yellow.

The present document lists all your comments and those from the reviewers, with our corresponding answers highlighted in bold. Line numbers referenced in our comments correspond to the “marked-up version” of the revised manuscript.

Anubis VEGA-RUA (Corresponding author)

 

EDITOR’S COMMENTS: 

I agree with the points raised by the reviewers and find the following points deserve your attention:

Lines 87-88 Not sure of what you mean by “interplay.”

Response: Here “interplay” means “relationship”. To give more clarity, the sentence was modified (line 90).

Lines 172-173 Repolarization is very fast (<sec). There is probably a confusion with "sensory adaptation" which is a distinct phenomena

Response: You are right. To clarify and avoid any misunderstanding, the sentence was modified: “Time interval between two stimulations was 40 s” (lines 194-196).

Lines 337-339 Was the pH of the solutions monitored? Addition of large amounts of organic acids are expected to decrease the pH, which could affect behavior.

Response: You are right. The pH modulation by addition of organic acids may influence oviposition. As you recommended, we measured the pH for all the compounds, alone or in blend at 100 ppm, because we estimated that pH differences with respect to UPW will be the stronger when the highest amount of organic acid is used. We also measured the pH for the solutions that influenced significantly the oviposition of gravid females (blend at 1 ppm and myristoleic acid at 10 ppm), and for a larval infusion at 100 larvae / 100 ml. Here are the values we obtained: 

Ultrapure water (UPW): 5.35

Isovaleric acid 100 ppm: 3.36

Myristic acid 100 ppm: 4.97

Myristoleic acid 10 ppm: 5.30

Myristoleic acid 100 ppm: 4.43

Pentadecanoic acid 100 ppm: 4.98

Blend of compounds 1 ppm: 5.94

Blend of compounds 100 ppm: 4.01

Larval holding water (LHW) at 100 larvae / 100 ml: 5.42

These values are not correlated with the oviposition preference that we obtained: 

- The solution containing pentadecanoic acid, highly preferred by females for oviposition, has the same pH as the one observed for myristic acid, while this latter does not influence oviposition.

- The two solutions containing either myristoleic acid or the blend of compounds, both dosed at 100 ppm, are the ones which deterred the more the oviposition. Even if these solutions are more acid than UPW, their pHs are higher than the one of isovaleric acid solution at 100 ppm, which did not significantly influence the oviposition.

- The solution containing myristoleic acid at 10 ppm is also deterrent for gravid females. This solution has the same pH as the one measured for UPW, clearly demonstrating than the acidity of the solution is not the factor that drives oviposition preference here.

- The solution containing the blend of compounds dosed at 1 ppm, interestingly, has a pH of 5.94, higher than the one of UPW.

- The water that held larvae (LHW) has also a similar pH than the one of UPW. Here again, this shows that the pH does not seem to influence the preference of gravid females.

In general, we do not seem to observe a relationship between acidification of solution and oviposition avoidance. We added information relative to pH and oviposition in the methods (lines 138-140) and we referred to our findings on the results section (lines 276-279) and our discussion (lines 412-415).

Line 404 Typing errors: electroantennography instead of elactro; performance instead of performence

Response: Changes were done according to your remark (line 468).

Line 406. Please give their names

Response: The names of the trainees were added to the “Acknowledgements” section (line 471).

 

REVIEWERS’ COMMENTS: 

REVIEWER #1:

In this study, Boullis et al., describe the effects of larval-associated carboxylic acids on the oviposition behavior and olfactory detection of gravid female Aedes aegypti. Using a dual choice-bioassay, they provide evidence that individual compounds have distinct effects and that some of these effects are dose-dependent. Moreover, they show that the odor blend is more attractive than the control at the lowest dose. Finally, antennal responses were tested using the EAG technique, showing response to isovaleric acid. In general, the data presented in this study are interesting and valuable but appear incomplete.

Strengths

The combination of sensory and behavioral techniques are suitable to explore the effect of carboxylic acids on oviposition. The authors’ rationale is based on published evidence that larvae, which release carboxylic acids, attract gravid females.

Response: We would like to thank you for this general positive comment.

Weaknesses

Although others have looked at the behavioral effect elicited by 1,000-7,800 larvae, showing more data points (5) for the OAI would be desirable to show a more convincing dose-response relationship in the context of the authors’ bioassay. Two of the three data points (5 and 20 larvae/100mL) shown are not different.

Response: The main goal of this study was to assess the influence of larvae-associated carboxylic acids on the oviposition of gravid Ae. aegypti females. As the oviposition response towards water holding conspecific larvae (LHW) was already demonstrated in several studies (see refs 16 to 20), we did not want to deepen this point. As mentioned on lines 168-169, the role of our experiment with the LHW at the three different densities was just to confirm the attractive effect of LWH in our experimental conditions, in order to use it as a good control (100 larvae / 100 ml is density frequently observed in the field) to challenge our most attractive synthetic solutions. Besides, by increasing the larval density, mortality can be observed (see article from “Walsh et al. 2011, J. Vector Ecol., 36, 300-307”; personal observations), generating compounds associated to larval decomposition which may bias the oviposition response of the gravid females. For these reasons, we did not think that is was relevant to increase too much the density in our study.

The results shown in Table 2 (no difference between compounds and LWH) can also be explained by the mixing of both headspaces within the 27L container.

Response: Of course, due to the size of the cages volatile plumes of compounds can be mixed. However, as three of the four carboxylic acids that we used are not detected by antennae, we considered that the choice of the females were driven more by stimulation (i.e. contact) than by attracting (at distance) cues.

In the case of the experiment presented in Table 2, we believe that the mixing of both headspaces is not substantial and do not significantly disturb the choice of the gravid females.

What was the composition of the gravid females used in the bioassay experiments? In other words, were all gravid females controlled for the time post-blood feeding? The longer females hold laying eggs the more likely they are to lay eggs in any moist surfaces. This could profoundly confound the experiment outcomes.

Response: All females, whatever the solutions tested, were blood-fed between 7 and 10 days after the emergence of adults. After the engorgement, only the fully-engorged females were conserved for the experiments. The oviposition tests were performed three days after blood feeding. To consolidate this point, the word “exactly” was added to the text (line 156).

Skip-oviposition (line 377) may also influence the outcomes of the behavioral experiments. It is hard to gauge this effect in the mosquito strain used here since the authors did not dissect the ovaries of the females after the experiments.

Response: By “skip-oviposition”, we were not referring to “delayed oviposition” or “egg retention”, but rather to the preference of Ae. aegypti females to deposit eggs in different breeding sites. However, we believe that this phenomenon does not impede to observe differences in the preference between two breeding sites in our experimental set-up. 

Minor comments

Replace x-axis legend with “Isovaleric acid (g)”. Is this a log [dose]?

Response: Indeed the x-axis represents a log [dose] for isovaleric acid. However, reviewer #3 requested to also present the others compounds in this figure. The figure was also moved from the supplementary files to the main text. According to all the remarks, the figure and its legend were modified (now figure 4).

Lines 24-25. “Only isovaleric acid elicited antennal response, suggesting that the other compounds may act as tactile cues.” The authors should discuss the possibility that the other odorants are detected by the maxillary palps or the proboscis.

Response: The maxillary palps can indeed be involved in the detection of these compounds. However, as the three compounds that do not elicit antennal response are “heavy” molecules (carboxylic acids with at least a C14 carbon chain), we can presume about the low volatility of the compounds, and thus consider them as tactile cues.

However, the eventual implication of maxillary palps was discussed, in the text rather than in the abstract (lines 352-354).

Reference 3, Italicize species name.

Response: The species name was italicized in the reference 3 (line 498).

Please provide more detailed description of the test cages.

Response: The commercial reference of test cage and the color of the oviposition bowls were provided in the Material & Methods section (lines 157-158).

 

REVIEWER #2:

- Abstract: The conclusion “Pentadecanoic acid and the blend of compounds are promising lures for ovitraps as they could compete with LWH.” This is not 100% accurate as two compounds within the blend function as deterrent. I would suggest to re-word that phrase.

Response: Indeed, some compounds act as deterrent/repellent when presented alone, but favor oviposition when presented in blend with other compounds. This notion was added to the discussion (lines 424-426). Also, the idea of dose was added in the abstract (lines 31). 

- Line 30-31: It states that the blend of compounds is a promising lure for ovitraps as they can compete with LHW. However, results showed that two of the compounds function as deterrent. Perhaps it is better to state that some of the compounds could function as lures given that they increased oviposition.

Response: As described in our answer of your previous comment, the abstract was modified according to your remarks (line 31), as long as the discussion (lines 424-426).

- Line 90-92: Why did you choose only these four compounds? Wang and colleagues (27) identified 8 compounds that were present in L4 larvae.

Response: We selected only these four compounds because there are the four major compounds identified in the L4 larvae chemical signature.

To improve clarity, we changed the word “main” by “major” (line 94).

- Line 98: Spelling mistake: feed instead of fed

Response: The correction was made in the text (line102).

- Line 104: Please specify the five localities. This information is important in case another group wants to re-do the experiments.

Response: The five cities where Ae. aegypti larvae were sampled are “Les Abymes”, “Pointe à Pitre”, “Deshaies”, “Saint François” and “Anse Bertrand”. These localities were specified in the text (lines 108-109).

- Line 109: Please specify the volume of blood that was used for feeding the mosquitoes.

Response: Several runs of experiments were performed to obtain all the dataset. Two Hemotek feeders were used for each feeding event. Each Hemotek feeder contained 2 ml of fresh blood. For each feeding event, the volume of blood was 4 ml. The total amount of blood used for all the experiment was estimated at about 50 ml.

This information was provided in the text (lines 114-115 & 117-118).

- Line 110: There is no information on how you determined that the females were gravid.

Response: After each blood meal, the whole cage was placed in a 6°C fridge facility to cold-anesthetize all the mosquitoes. The fully-engorged females were visually sorted. This information was added in the text (lines 116-117).

- Line 124: Please specify the final concentration in oviposition bowls.

Response: The final concentrations in the bowls were those presented in the text line 120: 0.1, 1, 10 and 100 ppm of compound.

We tried to improve the manuscript clarity by rearranging the structure of this paragraph (lines 126-137).

- Line 128: Please specify why you used these three different densities.

Response: The density of 20 larvae / 100 ml corresponds to the optimal rearing condition which is used in the lab for routine rearing (200 to 300 larvae / L, as specified line 113-114). We decided to use the density of 5 larvae / 100 ml as a low density, with presumably a lower larvae-associated chemical signal. The density of 100 larvae / 100 ml is considered here as a high density, meaning a more crowded condition (presumably reinforcing the chemical signal). We chose these three densities because they are expected to be discriminant in terms of chemical signal and then behavioral output of the females. Also, the higher density in this set-up does not induce detrimental effect of density on larval survival (i.e. no mortality was observed under this density) and it can be found in the field (personal observations).

This explanation was added in the text to explain our choices (lines 144-148).

- Line 138: Advice for any future oviposition assay: Use either a bigger cage or smaller oviposition vessels. The oviposition vessels were quite large and this can have an olfactory effect between test and control in such a small cage thus causing an effect on the results.

Response: Thank you for the advice, we will consider your remark for our future experiments.

- Line 141: 24h is a short amount of time for the females to choose an oviposition site and complete oviposition. If you have based this time period on a previous study please specify.

Response: The presence of eggs in the water affects the oviposition choice of females, through both visual and chemical cues (see ref [25] and “Allan et al. 1998, J. Med. Entomol, 35, 943-947”). Consequently, we decided to run the experiment for 24h to obtain a sufficient number of eggs laid, but also to avoid the influence of eggs on the oviposition choice. Also, the duration of oviposition bioassays in the study of Ong & Jaal was 22h (ref [25]) and 23h in the study of Allan et al. 1998. You are right when considering that 24h is a short time and does not allow females to lay all their eggs, but this duration seems enough to observe a preference in the oviposition site.

- Line 172: Please specify how often the filter was changed and new aliquots were added.

Response: For each odor puff, a new piece of clean filter paper was inserted in a new clean glass Pasteur pipette, and a new aliquot of solution was added on the paper. This information was added in the text (lines 193-194).

- Line 181: Why did you not have the same number of repetitions as in previous EAG experiment?

Response: These two experiments were independent. We do not have the same number of replicates because the number of available insects was different.

- Table 1: There is no significant difference between the densities 5 and 20, could you please discuss this further? Also the different between 20 and 100 is only approximately 180 eggs. How would you explain these results?

Response: We observed a significant preference for each of the three tested densities when compared to ultrapure water (see results of statistical analyses in table 1). But when these preferences were compared between them, we did not observe any significant differences between 5 and 20, or between 20 and 100 larvae. The only significant difference was observed between the lower and upper tested densities (see figure 1).We hypothesize that the quantity of chemical cues used by females to make their choice is not different enough between two successive densities in our experiments. 

This point was further argued in the discussion (lines 315-320).

- Line 219: “Myristoleic acid exhibited lower OAI compared to other compounds”. This sentence is unnecessary as it is clear that it is negative across all concentrations and also it is explained in Line 224.

Response: This sentence was removed from the text (line 253-255).

- Line 222: I don’t quite understand where the comparison between each compound and UPW is shown. There is no bar in Figure 1 that shows UPW (which I am assuming is the control).

Response: In all our experiments, we represented the data by using Oviposition Activity Index (OAI). Each OAI value is calculated by taking into consideration the number of eggs laid in both bowls: the one containing the test solution, and the control one (containing UPW). The formula to calculate this index is indicated in the Materials & Methods section (lines 211-213). With this representation, each data point (each dot in Fig. 1 and each bar in Fig. 2) represents the comparison between the test solution and the control one, as a mean value characterizing the choice of the females. The same kind of representation was used in others studies (see refs 25 and 32).

We tried to clarify by detailing a little bit more in the “statistical analyses” part of the manuscript (lines 211-218).

- Line 227: Another comparison is made between compound and UPW however, the data is not presented in Fig. 2

Response: The comparisons between test (compound) and control (UPW) solutions are represented by the different bars in the Fig. 2 (each bar represents one compound at one dose). When the difference between test and control solutions is significant, it is represented in Fig. 2 by using asterisks. For the data points mentioned in this sentence (i.e. isovaleric acid) (lines 261-263), there is no statistical differences between test and control solutions.

As we mentioned for your previous comment, more information about the OAI was added in the Materials & Methods (lines 211-218).

- Line 230-232: Why did you include the compounds that act as a deterrent in the blend? I would suggest to add another oviposition experiment where you remove these compounds and test the ones that gave a positive oviposition effect. Also add them according to respective ratios. This might give a more conclusive result and also strengthen the conclusion that they may be a good candidates as lures in the field.

Response: The reason why we included all the four tested compounds in the blend was to better mimic the larval chemical signature, even if some of them are repellent/deterrent when tested alone. The idea here was to evaluate if synergistic effect can appear.

As you proposed, we performed additional oviposition assays. We tested the blend dosed at 1 ppm and at 100 ppm without myristoleic acid (the most and only significant deterrent compound). We selected these two concentrations because of the results obtained with the “complete blend” at these dose.

- At 1 ppm, the “complete blend” was attractive with a mean (± SEM) OAI of 0.21 (± 0.04). Without myristoleic acid, the mean (± SEM) OAI was 0.18 (± 0.07) (quite similar value) and the statistical difference with control solution was not significant (paired t-test: p-value = 0.068).

- At 100 ppm, the “complete blend” containing myristoleic acid was strongly repulsive, showing a mean (± SEM) OAI of –0.65 (± 0.08). Without myristoleic acid, the mean OAI was still negative (meaning repulsive) but with a lesser incidence on the oviposition, with a mean (± SEM) OAI of –0.21 (± 0.07). However the oviposition response was still considered as statistically significant (paired t-test: p-value = 0.048).

We can interpret these results as the following: when the blend of compound is not strongly dosed (i.e. 1 ppm), the repulsive effect that we observed with myristoleic acid alone is counterbalanced by the attractancy of others compounds, especially pentadecanoic acid. However, when highly dosed, the “complete blend” has the same deterrent effect as those observed with the myristoleic acid. When this latter is removed from the blend, the oviposition response seems similar to the response to isovaleric acid. We can hypothesize that, when slightly dosed, the oviposition response to the blend is the result of a complex synergistic effect, whereas when highly dosed, this oviposition response is driven by the repulsive compounds that compose the blend.

These new results were added to the text, in the Materials & Methods (lines 129-132), Results (lines 268-276) and Discussion sections (lines 395-400 & 404-407).

- Line 236: How come recently engorged females were used? It takes females approximately 48hrs to process the nutrients present in the blood-meal, it is only after this they actively seek after an oviposition site.

Response: The physiological status of the females were the same in both behavioral and EAG experiments. In this sentence, “recently engorged females” meant “gravid females” (i.e. 3 days post blood meal). To remove any doubt, the sentence was modified (lines 281-282).

- Line 332: It is strange that myristic acid did not have any influence on oviposition even though it has clearly been shown in ref (23,44,45). Did you try the concentration that was tried in these previous studies?

Response: In the ref 23, 44 and 45, the most attractive concentrations of myristic acid were 1, 0.01, and 10 ppm, respectively. In our setup we tested concentrations from 0.1 to 100 ppm, but none of these influenced the oviposition behavior. We hypothesized this is the result of differences in experimental setup and laboratory conditions, mosquito strain or mosquito individual experience.

- Figure 1: The graph is not well explained (in the text or in the figure legend). Is my interpretation correct?: That a density of 5 is significant different from 100 but not 5 from 20 or 20 from 100? Please clarify the results. Perhaps use bars instead of a linear graph?

Response: Your conclusion is correct. The aim of this figure is to show the density-dependent response; hence we prefer to represent the data with dots and line, with a numerical ordination of the x-axis. We added a sentence into the legend of the figure 1 to improve the understanding (lines 620-621).

- Figure 2: Control is not presented in the graph. The main text keeps mentioning the comparison between compounds and UPW but this is not presented in the graph.

Response: As we present the OAI values (as for figure 1), each bar takes in consideration the number of eggs laid in the treatment bowl and the corresponding control bowl. 

We tried to clarify this point by adding more information in the Material & Methods part (lines 211-218).

- Figure 3: Females seem to give a stronger response towards isovaleric acid post- oviposition. Why do you think this is? Also is it significant?

Response: This difference may be a result of a less intense response toward the solvent for one or two replicates within the post-oviposition group, thus increasing the ratio between isovaleric acid and the solvent. However, this greater response to isovaleric acid for females after oviposition is not significantly different from those of females before oviposition. 

REVIEWER #3:

This study by Boullis et al. focuses on the responses of Aedes aegypti female mosquitoes to cues associated with the presence of conspecifics larvae. More specifically the authors focused on 4 acids previously identified as chemicals produced by larvae. The main aim of the study was to determine the valence of these chemicals for females before and after oviposition. Due to the several deadly pathogens that Ae. aegypti females can transmit, it is essential to explore new avenues for vector control and targeting oviposition behavior is highly relevant.

The paper is clear, well written and rich in references. Data analyses are overall well conducted. Yet, the quality of the figures could be improved. I particularly value the fact that the mosquitoes used for the experiments are from a recently established colony which reduces the risk of genetic drift. One of my main concerns is that the reader does not have access to all the data the authors are mentioning so it is somehow difficult to assess their results which unfortunately affects the manuscript quality.

Response: We would like to thank you for this general description. According to your comment, we added general information about the methodology used in the manuscript and its supporting display items, and we made accessible the raw data of this article .

Specific comments:

- L132: Did you observe mortality in your larvae / pupae groups? I am wondering if dead larvae would influence the water odor profile.

Response: We did not observe any dead larvae / pupae at these densities. However to give you some insights, we also have performed some preliminary odor sample studies, with more crowded larval densities. When mortality appeared (due to crowding), we detected the presence of volatile putrefactive sulfur-based compounds (characteristic of decomposition), meaning that dead larvae influence the odor profile of water.

- L137: what color were the bowls?

Response: The bowls are dark red. This information was added to the text (line 156). 

- L141: did you control for female size / weigh?

Response: The size and weight of females were not monitored during the experiment. However, female rearing was standardized as much as possible between each modality/replicate. First, larval density (200 to 300 larvae / L) and food provided during the rearing process were homogeneous between trays, and considered as optimal. Second, for each set of behavioral test (about 10 to 15 cages), the females were homogeneously distributed in each cage. The word “homogeneous” was added to the text (line 161).

- L155-156: please rephrase. Currently sounds like you tested UPW at 0.1, 1, 10 and 100 ppm.

Response: The sentence was rephrased to avoid any misunderstanding (line 177).

- L172: I suggest humidifying the airflow for acquiring EAG data.

Response: The continuous airstream was indeed humidified. The information was added in the text (line 193).

- L173: “amplified”: please provide a value here. Also were your data filtered?

Response: The signal was amplified 10 times. The data were filtered using the classical “filter setting” of the software. These details were added in the text (lines 196-198).

- L174: were the females starved from sucrose before performing the EAGs?

Response: No, the females were just picked-up from the cage, cold anesthetized and prepared for EAG tests.

- L180: why not testing post-oviposition females as well? What about mated females but not blood-fed? The physiological status is expected to influence responses to odorants.

Response: We decided to test only gravid females in the dose-response experiment because we did not obtain differences between these two groups of mosquitoes when the highest dose (100 µg) was presented. Also, gravid females were used instead of post-oviposition females in EAG to better compare with the results obtained in behavioral assays (indeed, the same physiological status was obtained in both experiments). This explanation was added to the main text (lines 204-208).

We also tested mated but non blood-fed females (at 6-9, 9-12, and 12-15 days post emergence) with the highest doses of each compound, but we did not observe significant differences between the different groups of mosquitoes. We decided to do not show these data here because it was not the aim of this article.

- L187: Student t tests: did you apply a Bonferroni correction for multiple comparisons?

Response: Independent paired Student t-test comparisons were made for each modality; it means that only simple comparisons were performed here. In this sense, no Bonferroni correction was necessary.

- L200: conducting an ANOVA with such a small sample size is not appropriate.

Response: The repeated-measures ANOVA has been replaced by a linear mixed-effects model with the individual (i.e. mosquito) as a random factor (lmer function in R). Also, as you requested to present others compounds in the dose-response experiment, the same analysis was applied to other compounds. The outcomes of these analyses are detailed in the manuscript (lines 231-234).

- L285: tetradecanoic acid = myristic acid. This should be mentioned in the introduction.

Response: This information was added earlier, in the abstract and the introduction (lines 20 & 95).

- Figure 2. The blend was tested for the OAI, showing an attraction at 1 ppm. Why not testing it with EAGs? It would be interesting to test the 4 concentrations used for these oviposition experiments (i.e., 0.1; 1; 10; 100 ppm). Indeed, the combination of chemicals might trigger a higher antennal response, as it has been shown in mosquitoes and many other insect species.

Response: Of course the combination of chemicals can have a synergistic effect on the antennal response. However, the EAG tests showed that only isovaleric acid is perceived by antennae. It is then assumable that when presented in group, the four compounds do not induce any synergistic effect (because here again only isovaleric acid is perceived by antennae).

- Figure 3. Please provide exemplar EAG traces for each condition. Given that the conditions are independent (gravid or after oviposition), a space should be added between the bars. It would be great to see the raw data and how responses to acids differ from responses to your positive control, octenol.

Response: The figure 3 was modified according to your remarks: an exemplar trace was provided for each compound, under each physiological status, as well as a trace for negative and positive controls. Also, the bars from the two different groups of mosquitoes were more spaced. The legend of the figure 3 was modified accordingly (lines 627 & 633-634).

The raw data were provided to better assess the differences between each compound, the solvent and the positive control. Also, the comparison of the response between the different acids and the positive control 1-octen-3-ol can be observed on the figure 4 (ex-figure S1) representing the dose-response for each tested compound.

- Figure S1 should be included in the main paper instead of being provided as supplementary information.

Moreover, for transparency, please include the data obtained for the 3 other tested acids along with the responses obtained for the positive control (octenol) and for the solvent. I suggest creating a panel highlighting all these data within one figure. Why not including dose response curves performed with females after oviposition as well? It would be interesting to see if the threshold of detection is affected by the physiological status of the females.

Response: The figure S1 was included in the main text of the manuscript, as the figure 4. The dose-responses for 1-octen-3-ol and others organic acids were also added on the same figure, as you requested. Statistical analyses were performed for positive control. The legend of the figure was adapted (lines 635-640).

However, as mentioned in our response to one of your previous comment, we did not performed dose-response with females after oviposition.

- “Contact cues” or “tactile cues” are mentioned several times in the paper. I would replace it with “taste cues” for more accuracy.

Response: As you suggested, the expressions “contact cues” and “tactile cues” were replaced by “taste cues”. 

- Please provide page numbers in your revised manuscript.

Response: Page numbers were provided in the two versions of the manuscript.

---

## [Decision Letter · Decision Letter 1]

11 Feb 2021

Behavioural and antennal responses of Aedes aegypti (L.) (Diptera: Culicidae) gravid females to chemical cues from conspecific larvae

PONE-D-20-33407R1

Dear Dr. Vega-Rúa,

We’re pleased to inform you that your manuscript has been judged scientifically suitable for publication (but please make the corrections suggested by one referee) and will be formally accepted for publication once it meets all outstanding technical requirements.

Kind regards,

Michel Renou, Ph.D

Academic Editor

PLOS ONE

Additional Editor Comments (optional):

Reviewers' comments:

Reviewer's Responses to Questions

**Comments to the Author**

1. If the authors have adequately addressed your comments raised in a previous round of review and you feel that this manuscript is now acceptable for publication, you may indicate that here to bypass the “Comments to the Author” section, enter your conflict of interest statement in the “Confidential to Editor” section, and submit your "Accept" recommendation.

Reviewer #2: All comments have been addressed

Reviewer #3: All comments have been addressed

2. Is the manuscript technically sound, and do the data support the conclusions?

Reviewer #2: Yes

Reviewer #3: Yes

3. Has the statistical analysis been performed appropriately and rigorously? 

Reviewer #2: N/A

Reviewer #3: Yes

4. Have the authors made all data underlying the findings in their manuscript fully available?

Reviewer #2: Yes

Reviewer #3: Yes

5. Is the manuscript presented in an intelligible fashion and written in standard English?

Reviewer #2: Yes

Reviewer #3: Yes

6. Review Comments to the Author

Reviewer #2: (No Response)

Reviewer #3: L196: please replace "Filter setting" by the actual value in Hz. Is it a low-pass, high pass?

Figure 3B: please add the y-axis title.

7. PLOS authors have the option to publish the peer review history of their article (what does this mean?). If published, this will include your full peer review and any attached files.

Reviewer #2: No

Reviewer #3: No

---

## [Editor Report · Acceptance letter]

15 Feb 2021

PONE-D-20-33407R1 

Behavioural and antennal responses of *Aedes aegypti* (l.) (Diptera: Culicidae) gravid females to chemical cues from conspecific larvae 

Dear Dr. Vega-Rúa:

I'm pleased to inform you that your manuscript has been deemed suitable for publication in PLOS ONE. Congratulations! Your manuscript is now with our production department. 

Kind regards, 

on behalf of

Dr Michel Renou 

Academic Editor

PLOS ONE